# Characterizing and Eliminating the Inbreeding Load

**DOI:** 10.3390/vetsci11010008

**Published:** 2023-12-22

**Authors:** István Nagy, Thi Anh Nguyen

**Affiliations:** Institute of Animal Sciences, Hungarian University of Agriculture and Life Sciences (MATE), Guba Sándor u. 40, 7400 Kaposvár, Hungary; nguyen.thi.anh@phd.uni-mate.hu

**Keywords:** purging, lethal equivalent, ancestral inbreeding, old inbreeding, new inbreeding, inbreeding–purging model, purging coefficient, purged inbreeding coefficient

## Abstract

**Simple Summary:**

Inbreeding, which is the mating of related animals, is a general phenomenon in closed populations. Because it is often associated with a decrease in performance, it has been studied extensively during the past 150 years. In this review, the authors cover the pedigree-based and genomic procedures that are available to characterize inbreeding load and determine whether the interaction between selection and inbreeding reduces the inbreeding load in a given population. The pedigree-based methods include ancestral inbreeding, the inbreeding–purging model, and expressed opportunity of purging. The various ancestral inbreeding coefficients show the proportion of the genome of an animal that is identical by descent to its ancestors or the part of the inbreeding coefficient that is identical by descent to its ancestors. The inbreeding–purging model calculates the purged inbreeding coefficient, which takes into account that the frequency of deleterious genes is decreased by the interaction of inbreeding and selection. Finally, the expressed opportunity of purging calculates the proportion of the reduction in the inbreeding depression due to purging. Relevant studies relying on genomic methods are also presented. According to the findings of the reviewed studies, it could be concluded that although the harmful effects could be eliminated under specific circumstances due to inbreeding-aided selection, the application of voluntary inbreeding cannot be recommended.

**Abstract:**

The authors evaluated the relevant literature related to purging, which is the interaction between selection and inbreeding in which the population may eliminate its inbreeding load at least partially. According to the relevant literature, the inbreeding load and the process of purging were evaluated via pedigree methods based on ancestral inbreeding, the inbreeding–purging model, and expressed opportunity of purging, along with genomic methods. Most ancestral inbreeding-related studies were performed in zoos, where only a small proportion of the studied populations show signs of purging. The inbreeding–purging model was developed with Drosophila, and it was used to evaluate different zoo ungulates and Pannon white rabbits. Purging was detected in both studies. The expressed opportunity of purging was applied in Jersey cattle and Pannon white rabbits. In the Jersey cattle, it had an effect of 12.6% for fitness, while in the Pannon white rabbits, the inbreeding load was between 40% and 80% of its original value. The genomic studies also signalled purging, but they also made it clear that, contrary to the detected purging, the evaluated populations still suffered from inbreeding depression. Therefore, especially for domesticated animals, it can be concluded that deliberate inbreeding with the purpose of generating purging is not advocated.

## 1. Introduction

Inbreeding is defined as the identity by descent probability at any given autosomal locus and results from the mating of related individuals [1]. As mentioned by Sonesson et al. [2], nowadays, the truncation selection on the BLUP (Best Linear Unbiased Prediction) breeding values [3] is a standard selection method, which, compared to phenotypic truncation selection, increases not only the selection response but also the rate of inbreeding. Inbreeding decreases the frequency of heterozygous individuals relative to a defined base population, and therefore, if directional dominance is present, a reduction in mean performance for the trait under consideration occurs [4]. The increase in homozygosity has negative effects on heterozygosity, which is known as an individual’s genetic diversity [5]. As a result, higher levels of inbreeding lead to lower genetic diversity and inbreeding depression [6,7], which are two of the seven main genetic issues in the field of conservation biology [8]. It raises the importance of measurement in individual inbreeding because with such information, breeding programs can be well designed to control the unfavourable effects on individual fitness and population dynamics due to inbreeding [7,9]. The level of inbreeding is measured by the inbreeding coefficient, which is the probability that the two genes at any locus in an individual are identical by descent [10]. In the conventional approach, the inbreeding coefficients and other diversity-related parameters are pedigree-based statistics [11]. In recent years, the advancement of technology in generating genome-wide sequencing and genotyping data has allowed for more accurate estimation levels of homozygosity in the population [12]; however, pedigree-based analysis is contended to keep its vital role in conservation management [11]. Inbreeding depression considerably affects the survival of the inbred population [13] and population size, and eventually, species extinction may happen, especially for populations that are already small in size and have a closed structure. However, inbreeding depression is not limited to fitness traits, and it may affect any trait of interest in the wild, in zoos, and in domesticated animal populations (e.g., [14,15,16,17,18,19,20]). In order to minimize the unfavourable effects of inbreeding, standard procedures [21,22,23,24] can be applied in order to maintain long-term sustainable livestock production in the future [25]. However, under various circumstances, inbreeding avoidance is simply impossible. Zoological gardens often encounter the issue of having a captive population originating from a small number of animals, and obtaining additional animals is not possible. A well-known example is the Speke’s gazelle (*Gazella spekei*) population at the St. Louis Zoo, which was founded in 1969 by one imported male and two imported females. In 1972, another female was imported, but since then, no other animals have been imported [26]. The details of the management and breeding of this population can be found in the paper by Read and Frueh [27]. Because of the small population size, the population experienced a severe inbreeding depression. However, Templeton and Read [26,28] developed an unorthodox breeding program based on the following steps: rapidly increasing the population size, choosing inbred animals with genes from as many different ancestors as possible to be parents, and producing inbred offspring with diverse genetic ancestry. Templeton and Read demonstrated that after applying this breeding program, the inbreeding load was halved in just a three-year period, so they concluded that inbreeding depression did not cause an unsolvable problem in the context of the long-term maintenance of a population in which inbreeding cannot be avoided. Although this conclusion has been debated by others and has generated an inspiring back-and-forth debate [29,30,31,32], the breeding program of Templeton and Read [26,28] was probably the first example in which the authors deliberately used the interaction between inbreeding and selection in order to eliminate inbreeding depression (i.e., decreasing the inbreeding load). This phenomenon is called purging (Hedrick and García-Dorado [33]), which has mostly been described in laboratory conditions so far [34,35,36]. However, the number of available studies is much lower in zoo biology (e.g., [37]) and very low in domesticated animals (e.g., [38,39]). Therefore, the objective of this review was to summarize the characteristics of purging, including the necessary parameters and methods, to investigate and demonstrate whether purging is likely to happen in a given population, as neither purging nor the related parameters are widely known in animal or veterinary science.

## 2. Parameters Characterizing Inbreeding Load and Purging

### 2.1. Pedigree-Based Parameters

According to Hedrick and García-Dorado, Ref. [33] the inbreeding load is the genetic damage that is concealed in heterozygosis in the population and would be expressed in a complete homozygote. Under some simplified assumptions (fitness multiplicative across unlinked non-epistatic loci), inbreeding load equals the rate at which fitness declines with increasing inbreeding in the absence of selection (roughly, the % of reduction in fitness expected from each 0.01 increase in F), while purging is the increased purifying selection facilitated by inbreeding that can reduce both the inbreeding load and the actual depression of fitness. The related parameters are listed in Appendix A. The characteristics of the conventional inbreeding coefficient (F) [40] are considered to be known. The likelihood of genetic death (ll) is also an inbreeding coefficient, but it is only slightly related to F. When one is zero, the other must also be zero. If a given ancestor is assumed to have one lethal gene (l) and leads through a separate and uncomplicated line of descent to each of the parents of a particular animal, then the value of (ll) will be half of the F value due to this same ancestor, since F measures homozygosity for L and for l, but (11) measures homozygosity for the latter only. However, if the lines of descent are not separate or if one or more of the intermediate ancestors is also inbred with respect of the given ancestor, then the value of (11) will be less than half that of F [41].

The inbreeding load of the population can be characterized by the number of lethal equivalents where if it decreases over time in a given population then it signals the possibility of purging [28].

Alternatively, based on the inbreeding coefficients of (F), (F_BAL_), (F_KAL_), (*A*_HC_) and (F_OLD_), the possibility of purging is indicated either by the positive interaction between (F) and (F_BAL_) [42] or by the significantly positive effect any of the following inbreeding coefficient: (F_BAL_), (F_KAL_), (*A*_HC_) and (F_OLD_) on the examined trait (i.e., the possibility of purging is indicated by means of ancestral inbreeding of any type [29,42,43,44]). 

The purging coefficient signals the possibility of purging when it is significantly different from zero. In that case, the purged inbreeding coefficient is significantly lower compared to the Wright inbreeding coefficient (i.e., purging is indicated by the so-called inbreeding–purging model [45]).

The expressed opportunity for purging (O_E_) is the potential for reduction in expressed load in the present generation as a consequence of having inbred ancestors. The procedure wants to express the reduction of expressed load because of purging as a fraction of expressed load when purging is absent [46].

### 2.2. Genomic Parameters

The proportion of the genome that is identical by descent (IBD_G_) is estimated using molecular genetic markers. Inbreeding is characterized by the parameter called F_ROH_ estimated as the proportion of the genome in runs of homozygosity (ROH), and it ranges from zero to one. Thus, F_ROH_ is directly proportional to IBD_G_ [12].

## 3. Software

### 3.1. Pedigree-Based Parameters

There are several software options available to calculate the conventional inbreeding coefficient (F) [47,48,49] based on the algorithm of Meuwissen and Luo, (1993) [50]. On the contrary, the other inbreeding coefficients (F_BAL_), (F_KAL_), (F_NEW_) and (*A*_HC_) can be estimated using the GRAIN software developed by Baumung et al. [43], which was recently updated by Doekes et al. [51]. This software is based on the stochastic method of “gene dropping” [52,53] where the number of the iterations used is generally 1,000,000. The proportion of identity by decent (IBD) loci out of all loci in an individual genome is considered as its inbreeding coefficient (F). GRAIN calculates the F as the total IBD loci out of all loci in an individual genome (i.e., 1,000,000 in our case) [43]. In order to calculate the ancestral inbreeding coefficient of Ballou [42], GRAIN tracks IBD events in the pedigree of an individual (flagging the alleles). The proportion of flagged alleles out of all alleles provides (F_BAL_). Similarly, the proportion of flagged alleles in IBD state gives (F_KAL_). From the method of calculation, it is evident that (F_KAL_) is smaller or equal with that of F. When F is zero, (F_KAL_) is also zero. Both (F_BAL_) and (F_KAL_) must have values between 0 and 1 and can be considered as probabilities [42]. For *A*_HC_, the number of all IDB events are calculated for every allele. Unlike the ancestral inbreeding coefficient, this parameter can exceed one [42]. The likelihood of genetic death (ll) can be calculated using the R script developed by Kennedy et al. [54].

The purging coefficient (d) and the purged inbreeding coefficient (g) can be calculated using PURGd [55] and PurgeR [56].

The lethal equivalent can be calculated using the R script developed by Hoeck et al. [57].

After the parameters (listed in Appendix A) are calculated, they must be evaluated, where the method depends on the characteristic of the evaluated trait. If the trait has a normal distribution, then the effect of inbreeding (F, F_BAL_, F_KAL_, *A*_HC_ and F_OLD_) is determined by running a breeding value estimation procedure (animal model) [3] where the inbreeding coefficients are treated as covariates. ASREML [58] is a widely used software for performing breeding value prediction. On the contrary, if the trait of interest is binomial, then generalized linear mixed models (GLMMs) have to be fitted using the lme4 or pedigreemm package in R [59,60]. If the purging coefficient and the purged inbreeding coefficients are used, then the non-linear regression method has to be used in order to find more accurate values of these coefficients (nls function, stats package of R).

### 3.2. Genomic Parameters

The genomic inbreeding coefficient (F_ROH_) is estimated using PLINK version 1.07 software [12].

## 4. Correlation among Inbreeding Coefficients

### 4.1. Pedigree-Based Parameters

There are only a few available studies [61,62,63,64] estimating correlation coefficients among the various inbreeding coefficients. Based on Spanish and Hungarian rabbit populations, Curik et al. [61] and Piles et al. [63] both reported very high (0.97–1.00) correlation coefficients between (F_BAL_) and (F_KAL_) while somewhat a lower estimate (0.77) was reported in a Hucul horse population [64]. Piles et al. [63] also reported the possible maximum correlation coefficients (1.0) between (*A*_HC_) and (F_BAL_) and (*A*_HC_) and (F_KAL_), while Posta et al. [64] reported lower coefficients especially between (*A*_HC_) and (F_KAL_) (0.77). The conventional inbreeding coefficient showed high (0.88–0.90) correlation coefficients in both studies with all of the different ancestral inbreeding coefficients ((*A*_HC_), (F_BAL_) and (F_KAL_)). On the contrary, Piles et al. [63] reported only medium correlation coefficients between the old inbreeding (F_OLD_) and the conventional (F) and the other ancestral type ((*A*_HC_), (F_BAL_) and (F_KAL_)) inbreeding coefficients, where the reported values ranged between 0.6 and 0.7, respectively. The estimates in German sheep breeds ranged between 0.55 and 0.73 between the conventional and ancestral inbreeding coefficients [62]. The lowest correlations (0.0–0.57) were found among the new and ancestral inbreeding coefficients [61,63,64], but it has to be emphasized that the calculation of the new inbreeding coefficient was different based on either [29] or [44] and consequently the estimated correlations were different in the various studies where Piles et al. [63] reported very low (0.0–0.2) values while Curik et al. [61] and Posta et al. [64] observed low to moderate (0.17–0,57) values. Similarly, the new and the conventional inbreeding coefficients showed different correlation coefficients in the two studies caused by the same reason as before. Again, the reported values of Piles et al. [63] were low (0.2) compared the other studies [61,64] where much higher values were reported (0.67–0.90). Altogether, the results indicate that the conventional, new, and ancestral inbreeding coefficients are measuring different population parameters. These findings are important because not all inbreeding is expected to be equally harmful. As demonstrated by Doekes et al. [65], inbreeding in recent generations was more harmful than inbreeding on distant generations for yield, fertility and udder health traits in Dutch Friesian cattle. Therefore, inbreeding depression can be best characterized by using (F_KAL_) rather than the conventional Wright inbreeding coefficient. In addition, according to Schäler et al. [5], due to the identification of ancestral inbreeding, it is possible to select individuals with simultaneously high classical and ancestral inbreeding coefficients and mate them with unrelated animals in order to achieve purging effects.

### 4.2. Genomic Parameters

The conventional inbreeding coefficients showed moderate correlation with F_ROH_ (0.48–0.60) [5,66], and a similar value was observed between F_ROH_ and F_BAL_ (0.49), but interestingly, no correlation was found between F_ROH_ and F_KAL_ (0.00) [5]. Schäler et al. [5] noted that the estimates of F_BAL_ showed the highest positive correlation with F_ROH_ according to all other genomic estimates. Hence, it can be assumed that the F_ROH_ with its chosen parameter adjustments may describe ancestral inbreeding better than other genomic coefficients.

## 5. Estimates of Lethal Equivalents in Populations of Different Species

### 5.1. Pedigree-Based Parameters

The reported values of the estimated lethal equivalents in different species are summarized in Table 1. The first study using the term lethal equivalent was that of Morton et al. [67] analysing human data collected by Tabah [68,69] from Catholic marriage dispensations issued during 1919–1925 in two French departments (Morbihan and Loir et Cher). Two-thirds of the families were contacted, and the married couples were sorted into the groups of first cousins, 1.5th cousins, second cousins and unrelated, respectively. The same information was obtained from town clerks for a control sample of unrelated parents married during the same period and selected without regard to fertility or medical history. After determining the conventional (F) [40], the authors calculated lethal equivalents (B) from the weighted regression on (F) of the natural logarithm of the number of survivors. From Table 1, it can be seen that the calculated (B) was larger in Morbihan than in Loir et Cher and also that the (B) was smaller in the earlier life history trait of stillbirth and neonatal death compared to infant and juvenile death. In another similar human study [70], the genealogy of the Habsburg dynasty was evaluated between 1450 to 1800 covering more than 4 000 individuals along more than 20 parent-offspring generations. The analysed period was split into two consecutive periods of 1450–1600 and 1600–1800, respectively. Mortality data were classified into two categories: infant deaths (deaths in the first year of life, excluding miscarriages and stillbirths) and child deaths (deaths between years 1–10). Only deaths attributable to natural causes were considered for the analysis. Miscarriages and stillbirths were not included in the analysis because the information on such early deaths is sometimes contradictory in the historical sources. The effect of inbreeding was evaluated both for the mother and for the progeny. The importance of treating both litter and dam inbreeding simultaneously was emphasized by Falconer [71], who noted that litter inbreeding might reduce the viability of embryos while dam inbreeding may have an effect on the fertility of the females. In this study, the inbreeding coefficient of the mother did not affect either evaluated trait. According to their results, the authors reported that the inbreeding load for child survival showed a statistically significant strong reduction corresponding to almost 80% of the inbreeding load for child survival. On the contrary, for infant survival, an opposite tendency was observed where the inbreeding load was increased and this increase was very close to being significant (*p* = 0.06). The authors concluded that the genetic basis of inbreeding depression was probably very different for infant and child survival. Nevertheless, Ceballos et al. [70] concluded that although the contribution of environmental effects to the reduction of inbreeding depression cannot be completely discarded, the reduction in inbreeding load for child survival detected in the Habsburg dynasty is in accordance with theoretical evidence from models of purging.

As already mentioned in the introduction section, the captive Speke’s gazelle population consisted of only a few imported animals; therefore, avoiding the mating of related animals was not possible [26]. This population, similarly to [70], also experienced a decrease in its inbreeding load in the life history trait of 30-day viability, which was almost halved in a three-year period. However, it is widely known that inbreeding depression is much stronger in harsh conditions [79]. Armbruster and Reed [79] reviewed the current literature on the relationship between the magnitude of inbreeding depression and environmental stress and calculated haploid lethal equivalents expressed under relatively benign and stressful conditions based on data from 34 studies. Inbreeding depression increases under stress in 76% of cases, although this increase was only significant in 48% of the studies considered. Estimates of lethal equivalents were significantly greater under stressful (mean = 1.45, median = 1.02) than relatively benign (mean = 0.85, median = 0.61) conditions. This amounted to an approximately 69% increase in inbreeding depression in a stressful vs. benign environment. According to Armbruster and Reed [79], if the environmental effects were improved during this period for the Speke’s gazelle zoo population [26], then it also could explain the observed decrease of (B). In studies related to different bird species [54,57,72,73,74,75], there was a clear tendency of magnitudes of the estimated lethal equivalents increasing with the age until survival being monitored. When survival rates were followed until 1–2 years or until recruitment, different authors [5,62,63,64,65,66] reported higher LE (B = 3.4–6.9) compared to the study of Gruber et al. [75] analysing the early life-stage trait of hatching rate (B = 0.17).

### 5.2. Genomic Parameters

The Soay sheep (*Ovis aries*), Drosophila (*Drosophila melanogaster*) and killer whale (*Orcinus orca*) studies also followed this trend, where the inbreeding load was much higher for a late life history trait [76,77,78]. These studies are also unique from the point of view of the applied methodology, namely that the inbreeding load was determined using genomic methods (F_ROH_) [76,77,78]. It has to be noted that the lethal equivalents estimated by the various authors presented in Table 1. are not necessarily comparable, as they used different calculation procedures. According to Kennedy et al. [54], using the unstandardized version of their model, the estimated effect of (F) on the survival rate at birth provides directly the number of lethal equivalents per gamete (B). Regarding the adequate methodology, Grueber et al. [80] developed a model where the standardized coefficients were used to calculate lethal equivalents applying a model averaging method [80]. Regarding the different link functions (log-link, logit-link), different authors came to different conclusion as to which of the two functions show superior characteristics from the aspect of bias and fit of the data [81,82,83,84].

## 6. Studies Signalling Purging Based on Ancestral Inbreeding or Inbreeding–Purging Model

### 6.1. Pedigree-Based Parameters

The studies that are likely to observe purging are summarized in Table 2. The first comprehensive studies analysing the effect of ancestral inbreeding were performed on zoo populations [42,85]. In these studies, many captive populations were evaluated, but only a small fraction of these showed the signs of purging (one out of 25 in [42] and 14 out of 119 in [85]. Interestingly, there were some contradicting results between these studies. A good example is the Sumatran tiger population, which was analysed in both studies but only showed the signs of purging in the study of Ballou [42]; however, when it was reanalysed by Boakes et al. [85], they found no signs of purging. The two studies partly used different models, as in [42] a model was defined where any fitness trait was possibly affected by the inbreeding coefficient of the litter and the inbreeding coefficient of the dam plus the interaction between the inbreeding coefficient and the ancestral inbreeding coefficient of the litter (year of birth was also included in the model). In the study of Boakes et al. [85], an alternative model was adopted from Boakes and Wang [86] where the interaction term was not used, but the inbreeding coefficient (F) of the litter and the dam and the ancestral inbreeding coefficient of the litter (F_BAL_) were included in the model separately. Based on computer simulations, the proposed model of Boakes and Wang [86] proved to be more advantageous in situations when inbreeding depression is caused by mildly deleterious alleles. Computer simulations indicated that purging is more likely to occur when deleterious mutations are of a large effect and when inbreeding occurs slowly and over many generations [87].

The rabbit studies [61,63] were highly suitable from this aspect as they covered 25–40 generations where the inbreeding was only slowly accumulated. The studies show that in the Pannon white rabbit breed [94], this slow increase in inbreeding level was mainly due to the applied circular mating system [98]. However, by 2017, the Pannon white rabbit population showed that more than 65% of the rabbits’ genome had already experienced inbreeding in previous generations, making it less susceptible to inbreeding depression (Figure 1) [61].

This value is substantially higher than that of any examined zoo population [85], where the populations with the highest (F_BAL_) were Addax (*Addax nasomaculatus*) and Przewalskii’s horses (*Equus ferus przewalskii*) having mean (F_BAL_) values of 49.5 and 54.6%, respectively. However, it is important to note that the Pannon white rabbit population has a considerably lower inbreeding coefficient (Figure 1a,b) than the zoo populations examined where the mean (F) values were 18.4 and 21%, respectively. It has to be emphasized that in [61], signs of purging were only detected between 1992 and 1997, where the litter inbreeding showed significant inbreeding depression on the survival of kits at birth while one of the ancestral inbreeding coefficients (F_KAL_) had a significantly positive effect. However, (F_BAL_) had no effect on the examined trait. Later, no signs of purging were detected, but inbreeding depression was also absent between 1997 and 2017, so it was concluded that the effects of new inbreeding involving several genes with large harmful effects were already purged between 1992 and 1997 [61]. The other rabbit study [63] was also not fully consistent with respect to purging indication since these authors found the positive effects of the old inbreeding only, (F_OLD_), only on the slaughter and weaning weights, respectively, while the other ancestral coefficients (F_BAL_, F_KAL_ and *A*_HC_) had no effect on either trait. In the cattle studies [38,39], Hinrichs et al. [38] found a significantly positive effect of (F_BAL_) (but nonsignificant (F_KAL_)) on birthweight signalling purging, but this finding is still less than favourable as increasing birthweight may cause problems with calving. In addition, when Hinrichs et al. [38] applied the original model of Ballou [42], a significant positive interaction was observed between (F) and (F_BAL_) for birth weight and for stillbirth as well, showing the possibility of purging for both traits. As for calving ease, no signs of purging were found by any model. In [39], the results of (F_BAL_) and (F_KAL_) were consistent and both had a positive effect on the milk and protein yield of an Irish Holstein-Friesian population, but they did not influence other analysed traits, such as fat yield, calving interval, age at first calving and survival. Other examples for the possibility of purging were reported in White Shorthair goat [90] and in Pura Raza Espanola mares [91] where positive ancestral (F_KAL_) inbreeding effects were found for milk production [90], age at first foaling in months, average interval between first and second foaling in months and average interval between foaling in months [91].

If the possibility of purging is to be determined based on genealogical information, then besides ancestral inbreeding the so-called inbreeding–purging model is the other approach that can be used. The background theory was developed by García-Dorado [99], then the theory was tested in laboratory experiments using *Drosophila melanogaster* with different effective population sizes (between 6 and 50) [92,93]. Both studies recorded purging coefficients greater than 0 (0.02–0.30) and they concluded that in order to show purging, the product of the effective population size and purging coefficient has to exceed 1, which implies that purging should be efficient for population sizes in the region of a few tens and larger, but purging might be inefficient against nonlethal deleterious alleles in smaller populations [92,93]. In the case of captive mammals, López-Cortegano et al. [45] evaluated the genealogy of different threatened ungulate species of the family Bovidae with different demographic histories: barbary sheep (*Ammotragus lervia*), Cuvier’s gazelle (*Gazella cuvieri*), dorcas gazelle (*G. dorcas*) and dama gazelle (*Nanger dama*). These populations had different population sizes ranging between 4 (barbary sheep) and 39 (Dorcas gazelle). The study estimated purging coefficients larger than zero for all species (ranging from 0.08 to 0.48), but these estimates were only significant for the Cuvier’s gazelle and dama gazelle. Consequently, the conventional and the purged inbreeding coefficients were clearly separated for these species (Figure 2).

The direct consequence of purging is fitness rebound, where after the initial decrease in fitness while removing the inbreeding depression, the fitness of the population increases (Figure 3). This process is most apparent for the dama gazelle. Interestingly, of the four evaluated populations, the smallest and the largest populations did not show signs of purging. The barbary sheep was too small and, in this case, drift overcame purging, while for the dorcas gazelle, purging detection would probably require more generations [45].

To the best of our knowledge, the only study available in domesticated animals is one [94] where the authors re-analysed the Pannon white rabbit data of Curik et al. [61]. The only other study where the ancestral inbreeding and inbreeding–purging model were applied for the same dataset was [100], claiming that the inbreeding–purging model had superior predictive characteristics compared to ancestral inbreeding in predicting the future fitness of the evaluated population. The results of [94] were very similar to that of [61] in finding purging signs only between 1992 and 1997 but not afterwards. Regarding the predicted fitness, it showed partial purging (Figure 4), which means that after a certain period, the fitness stabilized and did not show further decrease. It also confirms the conclusions of [61] that in the first period, genes with large effects were purged contrary to genes with mild effects. Using pedigree information, the expressed opportunity for purging is yet another method quantifying the decrease of the inbreeding load. In a Jersey-cattle-related study, the opportunity for purging was such as to reduce the autozygous frequency of alleles with strong effect on fitness by about 12.6%. With the level of inbreeding in Jersey cattle, the expressed genetic load in the current generation is reduced by about 12.6% because of ancestral inbreeding, provided the fitness of the homozygous allele is near zero [46]. Reporting on the survival at birth of Pannon white rabbits, Kövér et al. [94] found that the inbreeding load started to decrease only after 10 generations, and by the end of the analysed period, the inbreeding load was between 40% and 80% of its original value, meaning that the decrease in load was at least 20% of the original magnitude. This result was completely in accordance with the other procedures (ancestral inbreeding and inbreeding–purging model [61,94]) further strengthening the possibility of purging detection in this rabbit population.

### 6.2. Genomic Parameters

With the advancing genomic methods, it is no wonder that when there is no pedigree available, the existence of purging can also be determined using whole genome analysis [95,96,97]. These studies always compare different populations of the same species (i.e., small–isolated and large–connected Bengal tiger (*P. tigris tigris*) populations; [95]; island vs. mainland Kākāpō (*Strigops habroptila*) populations [96] and Iberian (*Lynx pardinus*) vs. Eurasian (*Lynx lynx*) lynx) populations [97]) in order to evaluate the differences in the frequency and genomic distribution of putatively deleterious genotypes among the different populations. 

## 7. Application Possibilities of Purging, Future Perspective

Although the purging phenomenon was investigated extensively, especially in zoo populations [42,85] only a small fraction of these populations showed signs of purging and the observed amount of the purged inbreeding load was usually not too large. Since inbreeding can fix harmful mutations, there is a general consensus in the field of animal breeding that intentional inbreeding should be avoided [25] when possible. However, in conservation genetics, several studies suggest that based on different breeding designs (e.g., circular sib mating), inbreeding may be beneficial due to purging [101,102,103]. However, the efficiency of inbred mating depends on the balance between the loss of diversity, the initial decrease of fitness and the reduction in inbreeding load [104]. Therefore, the so-called application of purging should be treated with caution [105].

## Figures and Tables

**Figure 1 vetsci-11-00008-f001:**
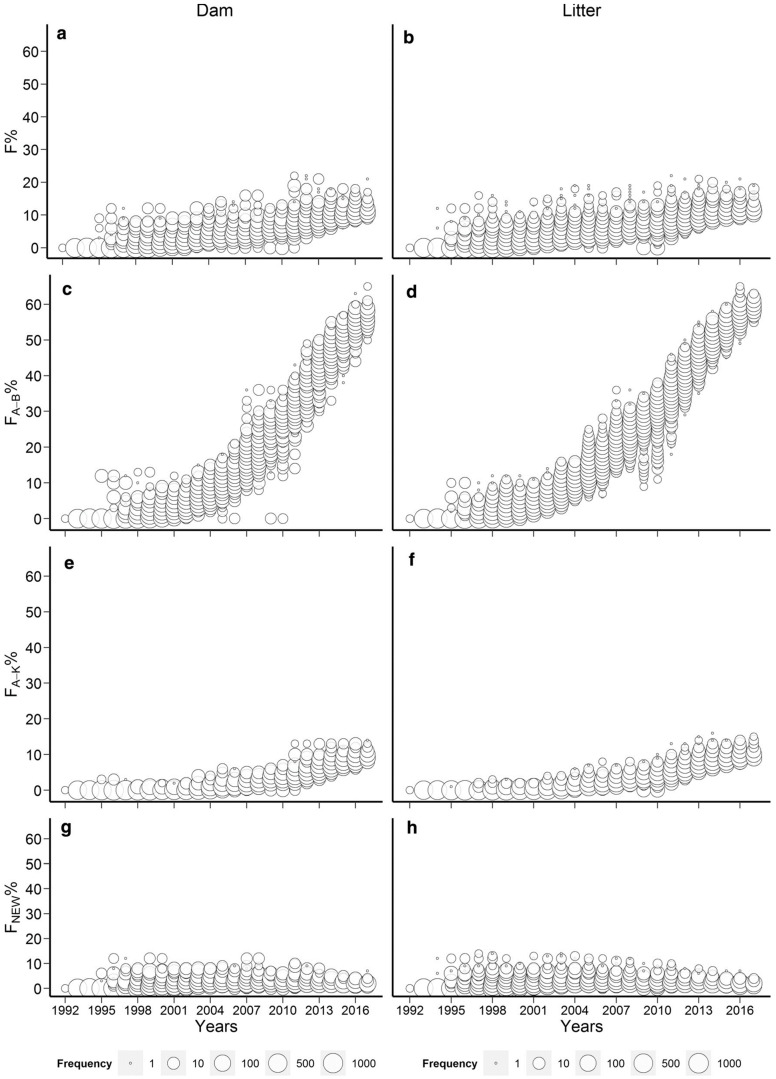
Evolution of the different dam and litter inbreeding coefficients of the Pannon white rabbits (Wright: **a**,**b**; Ballou: **c**,**d**; Kalinowski: **e**,**f**; Kalinowski new: **g**,**h**) [61].

**Figure 2 vetsci-11-00008-f002:**
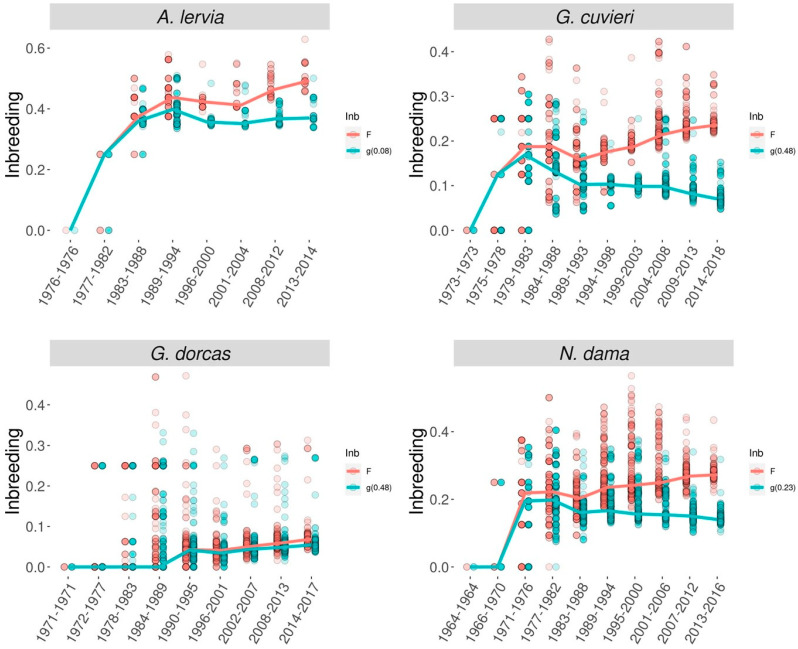
Evolution of the standard (*F*, red) and purged (*g*, green) inbreeding coefficients through time [45].

**Figure 3 vetsci-11-00008-f003:**
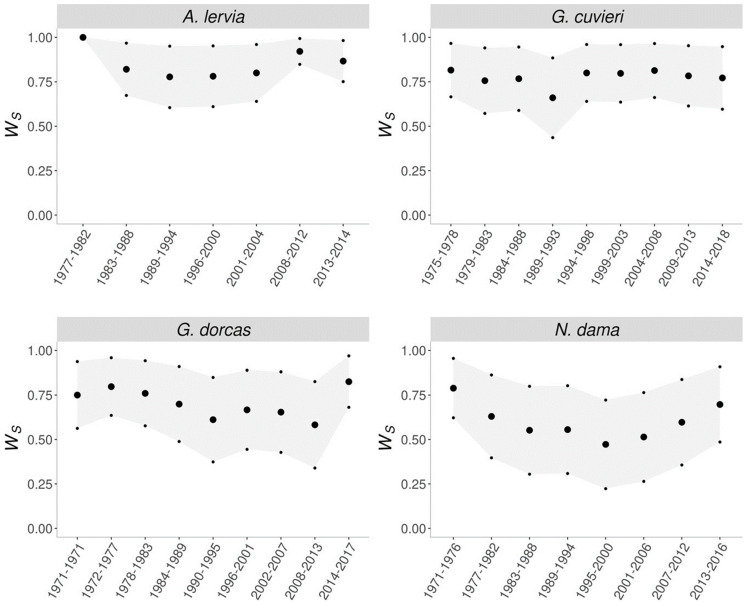
Evolution of the early survival [45] (large dots represent mean Ws, while small dots correspond to the mean value plus or minus one standard error).

**Figure 4 vetsci-11-00008-f004:**
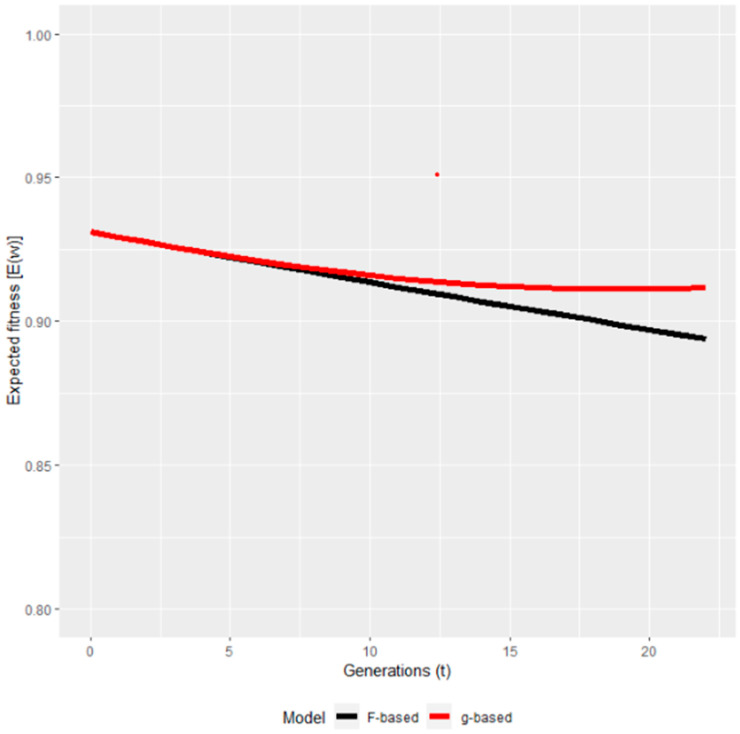
The predicted fitness based on the Wright (F) and on the purged (g) inbreeding coefficients [94].

**Table 1 vetsci-11-00008-t001:** Haploid lethal equivalents estimated in different species.

Species	Life History Trait	Estimated Lethal Equivalent	Reference
Homo sapiens	Stillbirth and neonatal birth	1.124 ^a^ 0.574 ^b^	[67]
	Infant and juvenile death	1.431 ^a^	
		0.908 ^b^	
Homo sapiens	Infant survival	0.396 ^c^	[70]
		2.745 ^d^	
	Survival to 10 years	4.373 ^c^	
		3.674 ^d^	
Speke’s Gazelle	30-day viability	2.97 ^e^	[26]
		1.59 ^f^	
North Island robin	Juvenile survival	4.14	[72]
Great tit	Survival to hatching	0.4	[73]
	Survival to fledging	0.4	
	Survival to recruitment	1.3	
	Survival to adulthood	2.12	
New Zealand robin	Juvenile survival	0.24	[74]
Takahe	Hatching rate	0.691	[75]
	Fledging rate	3.339	
	2-year survival	0.952	
	Offspring recruitment	3.383	
The Catham Island Black robin	Juvenile survival	3.42	[54]
Hawaiian crow	2-year survival	6.9	[57]
Soay sheep	1-year survival	2.285	[76]
*Drosophila melanogaster*	Total fitness	5.04	[77]
Killer whale	1-year survival	0.10 ^g^	[78]
		0.14 ^h^	
	40-year survival	2.74 ^g^	
		3.74 ^h^	

^a^ Morbihan; ^b^ Loir et Cher; ^c^ 1450–1600; ^d^ 1600–1800; ^e^ lethal equivalent of the imported animals; ^f^ lethal equivalent of the zoo population; ^g^ male; ^h^ female.

**Table 2 vetsci-11-00008-t002:** The observed purging cases in different species.

Species/Breeds	Analysed Trait	Used Methodology	Reference
German Holstein-Friesian	Birthweight	Ancestral inbreeding	[38]
Irish Holstein-Friesian	Milk yield	Ancestral inbreeding	[39]
	Protein yield	Ancestral inbreeding	[39]
Sumatran tiger	Neonatal survival rate	Ancestral inbreeding	[42]
*Gazella cuvieri*	Early survival	Inbreeding–purging model	[45]
*Nanger dama*	Early survival	Inbreeding–purging model	[45]
Jersey cattle	Fitness	Expressed opportunity for purging	[46]
Pannon white rabbit	Survival at birth	Ancestral inbreeding	[61]
Prat rabbit line	Weaning weight	Ancestral inbreeding	[63]
Prat rabbit line	Slaughter weight	Ancestral inbreeding	[63]
Amur tiger	Survival to 7 days	Ancestral inbreeding	[85]
Black-footed ferret	Survival to 7 days	Ancestral inbreeding	[85]
Lesser kudu	Survival to 7 days	Ancestral inbreeding	[85]
Grey dorcopsis wallaby	Survival to 30 days	Ancestral inbreeding	
Hippopotamus	Survival to 30 days	Ancestral inbreeding	
Congo peafowl	Survival to 30 days	Ancestral inbreeding	[85]
Black-footed ferret	Survival to 30 days	Ancestral inbreeding	[85]
Bontebok	Survival to 30 days	Ancestral inbreeding	[85]
Goeldi’s marmoset	Survival to 30 days	Ancestral inbreeding	[85]
Wied’s black-tufted-ear marmoset	Survival to 30 days	Ancestral inbreeding	[85]
Wyoming toad	Survival to 30 days	Ancestral inbreeding	[85]
Golden lion tamarin	Survival to 30 days	Ancestral inbreeding	[85]
Reindeer	Survival to 30 days	Ancestral inbreeding	[85]
Gunther’s dik-dik	Survival to 30 days	Ancestral inbreeding	[85]
*Peromyscus polionotus* *rhoadsi*	Litter size	Ancestral inbreeding	[88]
*Peromyscus polionotus* *rhoadsi*	Litter weight and weaning	Ancestral inbreeding	[88]
Border collie dog	Hip dysplasia	Ancestral inbreeding	[89]
White Shorthair goat	Milk production	Ancestral inbreeding	[90]
Pura Raza Espanola mares	AFF, I12, AIF	Ancestral inbreeding	[91]
*Drosophila melanogaster*	Egg to pupae viability	Inbreeding–purging model	[92]
*Drosophila melanogaster*	Non-competitive pupae productivity	Inbreeding–purging model	[93]
*Drosophila melanogaster*	Competitive productivity	Inbreeding–purging model	[93]
Pannon white rabbit	Survival at birth	Inbreeding–purging model	[94]
Pannon white rabbit	Survival at birth	Expressed opportunity for purging	[94]
Indian tiger	NA	Whole genome analysis	[95]
Kākāpō	NA	Whole genome analysis	[96]
Iberian lynx	NA	Whole genome analysis	[97]

AFF: age at first foaling in months; I12: average interval between first and second foaling in months; AIF: average interval between foaling in months.

## Data Availability

Data contained within this article.

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
