# Peer review of "Characterizing and Eliminating the Inbreeding Load"

_vetsci, 2023, doi:10.3390/vetsci11010008_

Round 1
Reviewer 1 Report
Comments and Suggestions for Authors
The review paper on inbreeding load is interesting. I would suggest to include more references about inbreeding depression in Livestock species. There are some studies about genomic inbreeding and also related to the inbreeding load that are interesting to add (e.g. https://doi.org/10.1186/s12711-020-00582-2, DOI: 10.1093/jas/skab316). The concept of inbreeding is not clear throughout the manuscript. It should be clarified.
Minor comments: I would revise the title of the manuscript, inbreeding load is equal to inbreeding depression?
Comments on the Quality of English LanguageSome spelling mistakes need to be corrected.
Author Response
Response to Reviewer 1
First of all I would like to thank the Reviewer the evaluation of the manuscript
„The review paper on inbreeding load is interesting. I would suggest to include more references about inbreeding depression in Livestock species.”
Athors are pleased that the manuscript is interesting. According to the request of the Reviewer, our work has been extended with references analyzing domesticates animals.
„There are some studies about genomic inbreeding and also related to the inbreeding load that are interesting to add (e.g. https://doi.org/10.1186/s12711-020-00582-2, DOI: 10.1093/jas/skab316).”
Thank you for these suggestions. The second reference is added to our work. The authors completely agree with the reviewer that these articles are both very interesting. Unfortunately, the first proposed article is based in the individual specific inbreeding load (developed by Casellas) which procedure is worth of veryhigh attention but we did not want to mention yet another procedure besides ancestral inbreeding, inbreeding-purging model, expressed opportunity of purging and genomic methods.
„The concept of inbreeding is not clear throughout the manuscript. It should be clarified.”
Authors regret that the inbreeding concept was not clear enough in the manuscript. Sincet he inbreeding is a general phenomenon in the closed population the main idea of our review was to summarize what different approaches were uses in the relevant literature to quantify the genetic load and how the reduction of the inbreeding load (due to purging) can be signalled.
Minor comments: I would revise the title of the manuscript, inbreeding load is equal to inbreeding depression?
„The title is modified according to the request”.
Reviewer 2 Report
Comments and Suggestions for Authors
The authors present a review on inbreeding depression, inbreeding load and purging effects in domestic animals, zoo and laboratory populations. The different approches and methods are described. The authors consider pedigree-based methods and genomic methods.
Comments
Some references particularly in regard to the ancestral inbreeding and lethal load are missing. The authors should search literature for newer papers.
Simple Summary: should be improved. Please give a short outline of the different approaches. The basic principle of calculating the inbreeding coefficient underlies also the different methods to derive ancestral inbreeding. Also opportunity of purging is based on the classical inbreeding coefficient.
The Abstract does not reflect the contents of the manuscript. The main outcomes of this review and the conclusions for further research have to be addressed. The authors are just telling the readers what they did but results and the main conclusions are missing.
Line 10-11: "..inbreeding depression is usually characterized as the regression coefficient of the examined trait’s 10 measurements on the inbreeding coefficients of the animals." >> as the linear regression coefficient
Line 94: there seems some misunderstanding of purge. With purging, the loss of fitness is less than expected from inbreeding due to selection against homozygous lethal variants.
Lines 96-136: this list may be summarized in a Supplementary Table and supplemented with comments. In addition, the formulas and some examples to explain the calculations would be useful. Software can also be added in this Suppl. Table.
The authors should split chapter 2 in two subchapters, the first part may contain pedigree-based methods and the second part genome-based methods. This subdivision should also be kept in the following sections.
Line 156: [59-59], please amend.
Line 160-180: there are references missing. This part contains a listing of results. What do these correlations tell us? Please explain the worth of these correlations.
Comments on the Quality of English Language
No comments
Author Response
Response to Reviewer 2
Authors would like to thank the Reviewer for evaluating our manuscript.
„The authors present a review on inbreeding depression, inbreeding load and purging effects in domestic animals, zoo and laboratory populations. The different approaches and methods are described. The authors consider pedigree-based methods and genomic methods.
Some references particularly in regard to the ancestral inbreeding and lethal load are missing. The authors should search literature for newer papers.”
Thank you for this suggestion. According to the request the manuscript has been extended with articles related to various livestock species from the recent past.
„Simple Summary: should be improved. Please give a short outline of the different approaches. The basic principle of calculating the inbreeding coefficient underlies also the different methods to derive ancestral inbreeding. Also opportunity of purging is based on the classical inbreeding coefficient.”
The simple summary has been rewritten completely. The expressed opportunity of purging is also covered.
„The Abstract does not reflect the contents of the manuscript. The main outcomes of this review and the conclusions for further research have to be addressed. The authors are just telling the readers what they did but results and the main conclusions are missing.”
The abstract has been rewritten completely.
„Line 10-11: "..inbreeding depression is usually characterized as the regression coefficient of the examined trait’s 10 measurements on the inbreeding coefficients of the animals." >> as the linear regression coefficient”
The text of the simple summary has been completely rewritten and this part has been deleted.
„Line 94: there seems some misunderstanding of purge. With purging, the loss of fitness is less than expected from inbreeding due to selection against homozygous lethal variants.”
This sentence has been corrected.
„Lines 96-136: this list may be summarized in a Supplementary Table and supplemented with comments. In addition, the formulas and some examples to explain the calculations would be useful. Software can also be added in this Suppl. Table.’
According to the suggestion these lines were moved to Supplementary table.
„The authors should split chapter 2 in two subchapters, the first part may contain pedigree-based methods and the second part genome-based methods. This subdivision should also be kept in the following sections.”
All chapter were clarified with Pedigree-based parameters and with genomic parameters
„Line 156: [59-59], please amend.”
Thank you for pointing out this error. It has been corrected.
„Line 160-180: there are references missing. This part contains a listing of results. What do these correlations tell us? Please explain the worth of these correlations.”
This section has been extended. Also some interpretation is given.